## Research Article

microplastics; Pacific Island fisheries; local knowledge; contamination; bioindicators

**Corresponding author:**
Amanda Kirsty Ford;
Email: amanda.ford@usp.ac.fj

# A community-informed microplastic exposure index based on Pacific Island fisheries: Integrating local knowledge with empirical data

Amanda Kirsty Ford[1] [ID], Salanieta Kitolelei[1,2], Rufino Varea[1,3], Brian Stockwell[1], Joycinette Vosumbe Botleng[1], June Brian Molitaviti[4], Semese Alefaio[5], Lotokufaki Paka Kaitu[5], Lavata Nivaga[6,7], Kelly Brown[1], Cherie Morris[1], Eseta Drova[1], Siutiti Fe'ao[6,8] and Jasha Dehm[1]

[1]Centre for Sustainable Futures, The University of the South Pacific, Suva, Fiji; [2]Leibniz Centre for Tropical Marine Research (ZMT) GmbH, Bremen, Germany; [3]Office of the Secretariat, Pacific Islands Climate Action Network, Fiji; [4]Vanuatu Fisheries Department, Port-Vila, Vanuatu; [5]Tuvalu Fisheries Authority, Tuvalu; [6]School of Agriculture, Geography, Environment, Ocean and Natural Sciences, The University of the South Pacific, Suva, Fiji; [7]Live and Learn, Tuvalu and [8]Graduate School of Bioresource and Bioenvironmental Sciences, Kyushu University, Fukuoka, Japan

## Abstract

Coastal fisheries are central to Pacific Island nutrition, livelihoods and cultural identity, yet growing microplastic contamination threatens food security and public health. This study integrates fishers' knowledge of locally important coastal fish species with empirical measurements of microplastic loads to identify priority taxa for monitoring across Fiji, Tonga, Tuvalu and Vanuatu. Interviews with 110 fishers documented commonly caught species, and the number of times each taxon was reported was calculated. Family-level catch data and mean microplastic loads were each standardised between 0 and 1 to generate Catch and Microplastic Scores, which were multiplied to create an Exposure Index reflecting both social relevance and contamination levels. Regionally, Lethrinidae and Scombridae had the highest Exposure Index values, while Acanthuridae, Lutjanidae, Scaridae and Serranidae emerged as country-specific priorities. Gendered fishing patterns revealed differences in catch, influencing potential exposure pathways and highlighting the need for gender-disaggregated data in future assessments. This approach of combining local knowledge with contamination studies offers a replicable, regionally-grounded method for identifying key indicator species for future microplastic monitoring. Species within the Lethrinidae family, particularly *Lethrinus harak*, stand out as regional priorities because of their importance to subsistence and artisanal fisheries, exposure to microplastics and consistent occurrence across the region.

## Impact statement

Pacific Island communities depend on coastal fisheries not only for daily nutrition and livelihoods, but also for cultural identity. However, these fisheries are impacted by a variety of anthropogenic stressors, including microplastics. As research is expanding in the Pacific Islands to fill geographical gaps in marine microplastic pollution, it is important that future research is designed to be locally and regionally relevant. This study provides the first regional assessment of how microplastic contamination intersects with the species that Pacific people catch most frequently. By combining interviews with 110 fishers across Fiji, Tonga, Tuvalu and Vanuatu with empirical analysis of microplastic loads in commonly eaten fish, we developed an Exposure Index that ranks species according to both their importance to local diets and their measured contamination levels. This innovative approach moves beyond simply documenting pollution by identifying the fish families that warrant priority attention for monitoring and provides a framework that could be applied to future management and potential public-health guidance. We foresee three main impacts of this study. First, it demonstrates how integrating local fisher knowledge strengthens scientific relevance, ensuring that research outputs are meaningful to the communities most affected. Secondly, it equips Pacific governments, regional agencies and researchers with a practical, evidence-based tool for prioritising monitoring efforts. Finally, the framework is transferable to other island nations and coastal regions worldwide that face similar challenges of limited resources, high seafood reliance, and rising plastic pollution.

## Introduction

Coastal fisheries resources are crucial for communities in the Pacific Island Countries and Territories (PICTs), providing not only a vital source of nutrition but also sustaining livelihoods,

economies and cultural traditions that have endured for generations. While fisheries are critically important for food security, with people in PICTs consuming 3–6 times more fish than the global average (Bell et al., 2009), fisheries represent far more than just food security across the region (Johannes, 1981; Kitolelei et al., 2021). However, these critical resources face unprecedented threats including overfishing, poaching, destructive fishing methods, climate change and pollution (Pratchett et al., 2011; Jupiter et al., 2014). Among these challenges, the spread of plastics throughout marine systems presents complex risks that intersect environmental health and human nutrition.

PICTs, despite contributing around 1% of global plastic waste (Jambeck et al., 2015), face a disproportionately high burden of marine plastic pollution (Filho et al., 2019), due to their relatively small land areas; a threat of which is further emphasised by the role of plastics as a threat (e.g. climate change, biodiversity loss) multiplier (Costa, 2025). The ubiquitous presence of plastics in the ocean creates risks for marine wildlife and the environment through ingestion, habitat degradation, bioaccumulation, suffocation, strangulation and entanglement (Wright et al., 2013; Welden, 2020; Tekman et al., 2022). Fish, as well as other marine organisms, are highly susceptible to entanglement from plastics and discarded fishing gear (Nelms et al., 2016; Ryan, 2018; Jepsen and de Bruyn, 2019), particularly given the transboundary movement of plastic debris and the wide-ranging movements and feeding behaviour of many species.

Microplastic pollution poses an emerging threat to marine ecosystems and food security (De-la-Torre, 2020; Kibria, 2023). These microplastics, defined as particles larger than 1 μm and smaller than 5 mm, originate from both the breakdown of larger debris and direct inputs like microbeads from personal care products or synthetic fibres from clothing (Cole et al., 2011; GESAMP, 2015; Frias and Nash, 2019). What makes microplastics particularly concerning is their ability to infiltrate marine ecosystems. A global review found 49% of fish show evidence of microplastic ingestion (Wootton et al., 2021b). Microplastics enter the food web through ingestion by lower trophic level marine organisms, eventually reaching higher trophic level organisms, including potentially humans (Mercogliano et al., 2020; Lehel and Murphy, 2021). The health implications are potentially severe; microplastics not only pose physical hazards but also act as carriers for toxic chemical additives and environmental pollutants (Andrady, 2011; Koelmans et al., 2014). Plastic polymers contain concerning additives like phthalates, bisphenols and flame retardants, while their porous surfaces efficiently absorb persistent organic pollutants and heavy metals from seawater, including known carcinogens like benzo[a]pyrene and polychlorinated biphenyls (Kinigopoulou et al., 2022; Okoye et al., 2022).

In the context of Pacific Island fisheries, this contamination pathway becomes especially alarming as microplastics are ingested by coastal fish species that form the cornerstone of local diets (Wootton et al., 2021a; Drova et al., 2025). Fish consumption in the PICTs is exceptionally high, with coastal rural populations consuming between 30 and 146 kg of fish per person annually, vastly surpassing the global average of 16–18 kg even in many urban centres (Bell et al., 2011). Many coastal communities rely on seafood as a primary food source, often consuming fish in ways that utilise the whole animal; a practice rooted in both tradition and food security. There is now concrete evidence of the prevalence of microplastics in coastal areas (Bakir et al., 2020; Dehm et al., 2020; Ferreira et al., 2020; Markic et al., 2022, 2023), including several recent studies on important food fish (Botleng et al., 2025; Drova

et al., 2025; Fe'ao et al., 2025; Alefaio et al., 2026). Yet despite these growing concerns, critical knowledge gaps persist regarding the threat to Pacific Island food systems and coastal communities. Furthermore, as research in this field expands, it becomes increasingly relevant to target future research on suitable indicator species at a larger scale. Most scientific studies focus on temperate regions or commercial seafood species (Wootton et al., 2021b), overlooking the unique species compositions and consumption patterns of subsistence fishers from Pacific communities. Furthermore, existing research often fails to incorporate local fisher knowledge that could reveal important exposure pathways.

Understanding the risk of microplastic exposure in rural communities requires a comprehensive approach that integrates both scientific data and local fisher knowledge. Small-scale fishers rely on knowledge built over generations regarding fish species and their habitats, seasonal variations and feeding behaviours (Johannes et al., 2000; Stephenson et al., 2016). They have invaluable insights into the species present in their fishing grounds, catch composition, as well as differences in catch between male and female fishers (Kitolelei et al., 2021). While quantitative science-based approaches can document microplastic ingestion in marine organisms, integrating fisher knowledge provides critical context on the species that are most frequently caught and consumed. Such local knowledge can significantly enhance scientific assessments of microplastic contamination in marine food webs, highlighting that fishers should be recognised as valuable partners if we want to provide relevant and impactful assessments of microplastic exposure risk. Such collaborative approaches can strengthen risk assessments and inform more effective mitigation strategies.

This study addresses these gaps by integrating fisher knowledge with empirical microplastic data to quantify exposure risks in four Pacific Island nations. Our approach recognises that fishers possess unparalleled insights into species availability and consumption patterns; knowledge essential for accurate risk assessment and prioritising indicator species. By documenting community-caught fish species and linking this to data on taxon-specific fish microplastic loads, this study aims to improve our understanding of microplastic fish-human pathways. The objectives were to (1) document (gender-disaggregated) fisher knowledge of coastal fish landings to rank fish by country and region and (2) link these catch data to microplastic contamination levels to develop a community-informed Exposure Index. Through our approach, which prioritises fisher-reported catch data, we consider the species that matter most to Pacific communities and provide locally guided recommendations for targeting future microplastics research.

## Materials and methods

### Study sites

This study focuses on four Pacific Island countries: Fiji, Tonga, Tuvalu and Vanuatu (Figure 1). In each country, three rural coastal fishing communities were chosen based on their geographic location, relevance to regional fisheries partners, and dependency on coastal fisheries for subsistence. These selected sites offer an overview of how exposure risk may vary across subsistence fisheries with distinct catch compositions and exposure pathways. Data collection dates fell between July 2023 and November 2024 (Supplementary Table 1).

The selected study communities from Fiji were Galoa (Serua Province), Yadua (Nadroga-Navosa Province) and Silana (Tailevu Province); all located around Fiji's main island Viti Levu. Sites were

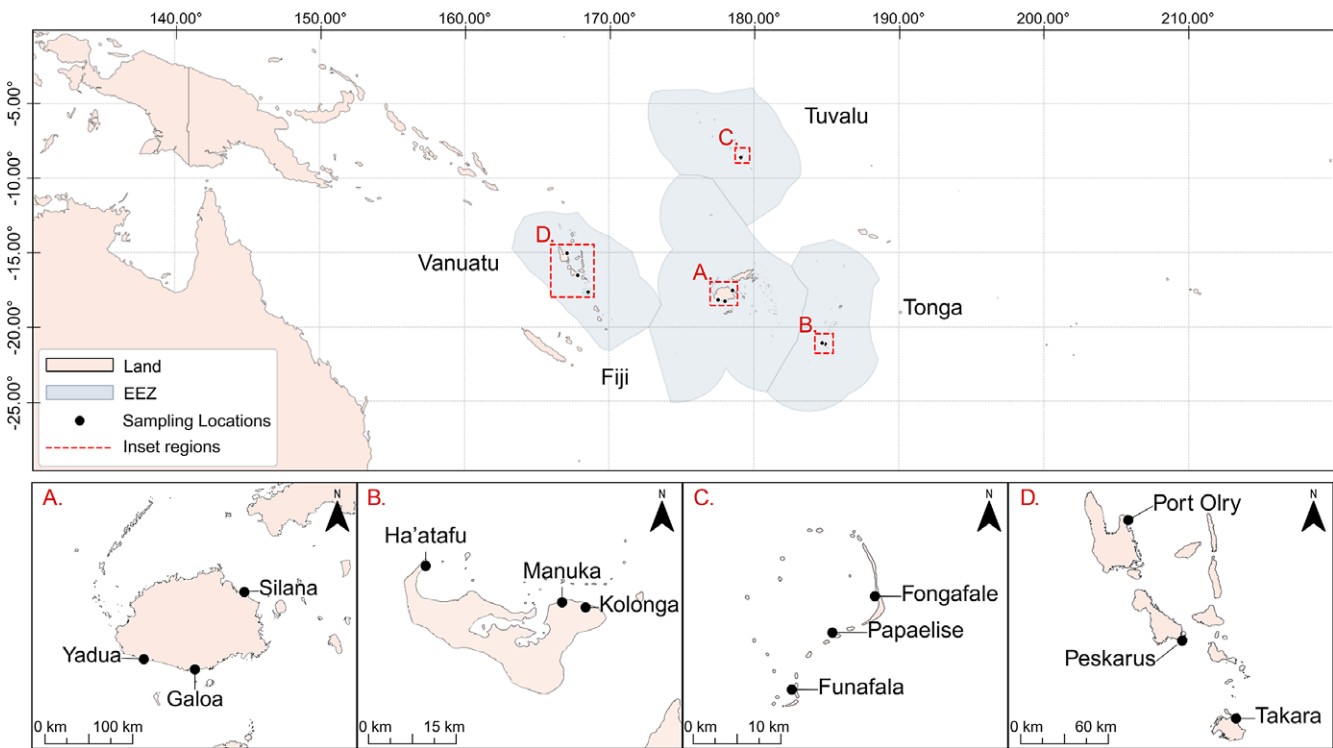

**Figure 1.** Coastal communities where fishing practices and species landings were documented in collaboration with local fishers: (A) Fiji's Viti Levu island (Villages of Galoa, Silana and Yadua), (B) Tongatapu, Tonga (Villages of Ha'atafu, Manuka and Kolonga), (C) Funafuti Atoll, Tuvalu (Islets of Fongafale, Papaelise and Funafala), Vanuatu's Efate (Takara), Malekula (Peskarus) and Espiritu Santo (Port Olry).

intentionally selected to be away from the capital Suva, which is already well-studied, to avoid the urban influence, which is at a scale unmatched by urban areas in the other countries. In Tonga, surveys were conducted in three villages on Tongatapu Island: Manuka, Kolonga and Ha'atafu. All villages have Special Management Areas in place as initiatives of the Ministry of Fisheries to enhance data collection and monitoring capabilities. In Tuvalu, surveys focused on three islets – Fongafale, Papaelise and Funafala – around the capital atoll Funafuti. In Vanuatu, surveys were conducted at Port Olry (Espiritu Santo), Peskarus (Maskelyne Island) and Takara Village A (Efate Island), in conjunction with creel surveys being conducted by the Fisheries Department.

### Documenting fisher knowledge

The data for this study were collected through questionnaire-guided interviews. In each country, an average of 28 interviews were conducted, with 7–10 questionnaires administered in each of the three selected communities, resulting in a total of 110 fishers (see Supplementary Table 1 for breakdown of men and women per country). The questionnaire formed part of a broader study documenting perceptions of plastic pollution sources, impacts and responses (see Kitolelei et al. in revision). However, for the purposes of this study, we analysed responses to a single open-ended free-listing question: '*What are the ten (10) main food fish caught in your fishing ground?*' Fishers were not prompted with a pre-constructed list or visual aids. To promote inclusivity and comfort, men and women were interviewed separately and according to their availability. The interviews adhered to principles of Free, Prior and Informed Consent (FPIC), and interviewees were encouraged to withdraw at any time if they no longer wished to participate. The

interviews were conducted in a culturally sensitive manner and translated into the local language.

### Microplastic sampling

Data on microplastic loads in commonly caught fish species were compiled from concurrent analyses conducted by the study research team across the four study countries (Fiji: Drova et al., 2025; Tuvalu: Alefaio et al., 2026; Vanuatu: Botleng et al., 2025; Tonga: Fe'ao et al., 2025), which all used consistent methodologies, allowing for comparability. In summary, at each study site, fish landed by local fishers were identified to the species level by the research team, who cleaned the fish externally, measured the fork length and total weight of each specimen and carefully isolated the gastrointestinal tracts (GITs) for storage (frozen) until subsequent analysis. Under laboratory conditions, GITs were digested using 30% $H_2O_2$ at a volume three times that of the GIT and incubated at 60 °C until organic tissues were fully dissolved. The digested material was then passed through a sieve series (500, 250, 125 μm; i.e. minimum measured particle size = 125 μm) and retained residue was visually examined via microscopy (100× magnification) to identify and categorise microplastics by form and size. For each dataset, a subset of the visually identified microplastics was validated using Fourier transform infrared spectroscopy (FTIR) polymer analysis following the methods described by Jung et al. (2018). Spectra were collected over the 4,000–450 $cm^{-1}$ range at 2 $cm^{-1}$ resolution and manually compared with reference spectra listed in Jung et al. (2018). Positive polymer identification required confirmation of at least four diagnostic absorption bands. To ensure quality assurance and quality control, all digestion and analysis procedures were conducted under controlled

laboratory conditions in batches of up to 30 samples. All glassware and equipment were thoroughly rinsed with filtered distilled water prior to use. Samples and solutions were covered with aluminium foil whenever possible to minimise airborne contamination, and exposure to open air was reduced during processing. Procedural blank controls consisting of open, sterile glass beakers containing 30% $H_2O_2$ were included with each digestion batch and processed identically to samples through digestion, sieving and microscopic analysis. For a complete list of species sampled for microplastics within each family, see Supplementary Table 2.

### Data analysis

To ensure sufficient replication for microplastics data among groupings, and due to some fishers providing fish information on a coarse taxonomic level (e.g. 'parrotfish' - see Supplementary Table 3), fish were grouped into their respective families for analysis. To visualise the most commonly caught fish families across countries, reported fish catch data were analysed and plotted using R (version 4.3.2; R Core Team, 2024). Relative contributions (as percentages) of the top five families were calculated per country, and plotted using *ggplot2* (Wickham, 2016). We then conducted a Principal Components Analysis (PCA) to further examine differences in fish family composition across countries. The analysis was based on the percentage contribution of the top five families per country (yielding a total of eight families overall). Country-level data were standardised by expressing family counts as relative percentages of total catch. The PCA was performed on the resulting dataset to identify patterns in community composition. A biplot was generated, displaying countries as points and fish families as vectors representing their contribution to the first two principal components.

For the calculation of the Exposure Index, fish were analysed first within each respective country, and then were compiled and analysed at a regional level. Families sampled for microplastics but not mentioned by fishers, or mentioned by fishers but not sampled for microplastics, were excluded from this calculation. Then, families identified as being commonly caught within fishing grounds were filtered to remove those without sufficient microplastic data ($n \geq 3$; see final country level replication in Supplementary Table 4, and regional level replication in Supplementary Table 5), resulting in the following number of families for each country: Fiji $n = 9$, Tonga $n = 10$, Tuvalu $n = 9$ and Vanuatu $n = 14$ families. The values for catch (defined as the frequency with which a fish belonging to a given family was identified in interviews as being caught within fishing grounds) were then standardised between 0 and 1, reflecting the minimum and maximum observed frequencies, respectively. This standardised metric is hereafter referred to as the 'Catch Score'. The same standardisation was conducted for microplastic loads (mean number of particles per individual within a family), hereon 'MP Score'. The product of these two separate metrics was calculated to provide an 'Exposure Index', with higher values reflecting combined importance in the catch and microplastic load (see Supplementary Figure 1).

To examine gender-based differences in fish catch composition, community-reported data were analysed using R with the *tidyverse* (Wickham et al., 2019) and *ggplot2* packages. Catch counts were grouped by country, gender and fish family using *dplyr* (Wickham et al., 2023), and proportional catch composition was calculated by dividing the number of fish caught by each gender by the total catch for that family. The gendered difference was defined as the proportion of the catch caught by women minus that caught by men.

Results were visualised using a faceted horizontal bar chart created with *ggplot2*, where positive values indicated families more commonly caught by women and negative values those caught by men.

### Results

A total of 110 questionnaire-guided interviews were conducted in 12 communities across the four countries. Overall, 60 % of the interviewees were men and 40% were women (Supplementary Table 1).

The most commonly caught families identified by fishers as being caught within their fishing grounds can be seen in Figure 2. Fishers in Fiji identified Lethrinidae to be their primary catch, followed by Serranidae, Carangidae, Lutjanidae and Scombidae. In Tonga, Acanthuridae were most commonly identified, followed by Scaridae, Siganidae, Scombridae and Lethrinidae. In Tuvalu, Acanthuridae dominated, followed by Lutjanidae, Serranidae, Carangidae and Lethrinidae. Finally, in Vanuatu, Siganidae were identified most commonly, followed by Lethrinidae, Scaridae, Acanthuridae and Lutjanidae. The differences across countries driven by the eight families that were found to be in the top five per country can be further visualised in the PCA biplot (Supplementary Figure 2).

Scores for catch, microplastics and Exposure Index for each family in each country can be seen in Table 1. For Fiji, the top three highest ranked families in terms of catch were Lethrinidae (38 mentions), Serranidae (27 mentions) and Lutjanidae (23 mentions). The highest ranked families in terms of microplastic (MP) loads were Scombridae (5.7 ± 1.2 MP/individual; mean ± SE), and then Serranidae (3.0 ± 2.1 MP/individual) and Lutjanidae (3.0 ± 1.4 MP/individual). Note that Terapontidae also had high levels of MP (3.5 ± 0.6 MP/individual), but as no fishers identified them as being caught, they are not included. The top-scoring families by the Exposure Index in Fiji were Scombridae, Serranidae, Lethrinidae (2.3 ± 0.2 MP/individual) and Lutjanidae. In Tonga, the highest three-ranked families by catch were Acanthuridae (17 mentions), Scaridae (14 mentions), Lethrinidae and Siganidae (both with 11 mentions). The highest ranked families in terms of MP loads were Carangidae (3.4 ± 1.3 MP/individual), Lethrinidae (0.9 ± 0.2 MP/individual), Mugilidae (0.8 ± 0.2 MP/individual) and Siganidae (0.8 ± 0.3 MP/individual). The families with the highest calculated Exposure Index in Tonga were Siganidae and Scaridae (0.5 ± 0.2 MP/individual), but values were low (≤ 0.12). For Tuvalu, the highest catch rankings were for Acanthuridae (49 mentions), Lutjanidae (38 mentions), Serranidae (36 mentions) and Lethrinidae. The highest microplastic loads were found in Siganidae (1.3 ± 0.8 MP/individual), Mullidae (1.2 ± 0.4 MP/individual) and Holocentridae (1.2 ± 0.5 MP/individual). The families with the highest Exposure Index values identified in Tuvalu were Acanthuridae (1.0 ± 0.3 MP/individual) and Lutjanidae (1 ± 0.6 MP/individual), scoring the highest across all countries (0.74 and 0.56 respectively). In Vanuatu, the top families ranked highest for catch were Siganidae (38 mentions), Lethrinidae (36 mentions), Scaridae (29 mentions) and Acanthuridae (28 mentions). All species had <0.4 MP/individual on average, but the families with the highest Exposure Index scores were Mullidae (0.13 ± 0.09 MP/individual), Lethrinidae (0.06 ± 0.05 MP/individual) and Scaridae (0.07 ± 0.05 MP/individual).

Clear gender-based differences are evident in reported catch for taxa where there were associated microplastics data (Figure 3). For those families with higher (threshold of 0.1) Exposure Index scores, in Fiji, women commonly reported Lutjanidae, while men reported Serranidae and Scombridae, and both genders equally reported Lethrinidae. In Tonga, men more frequently reported all families.

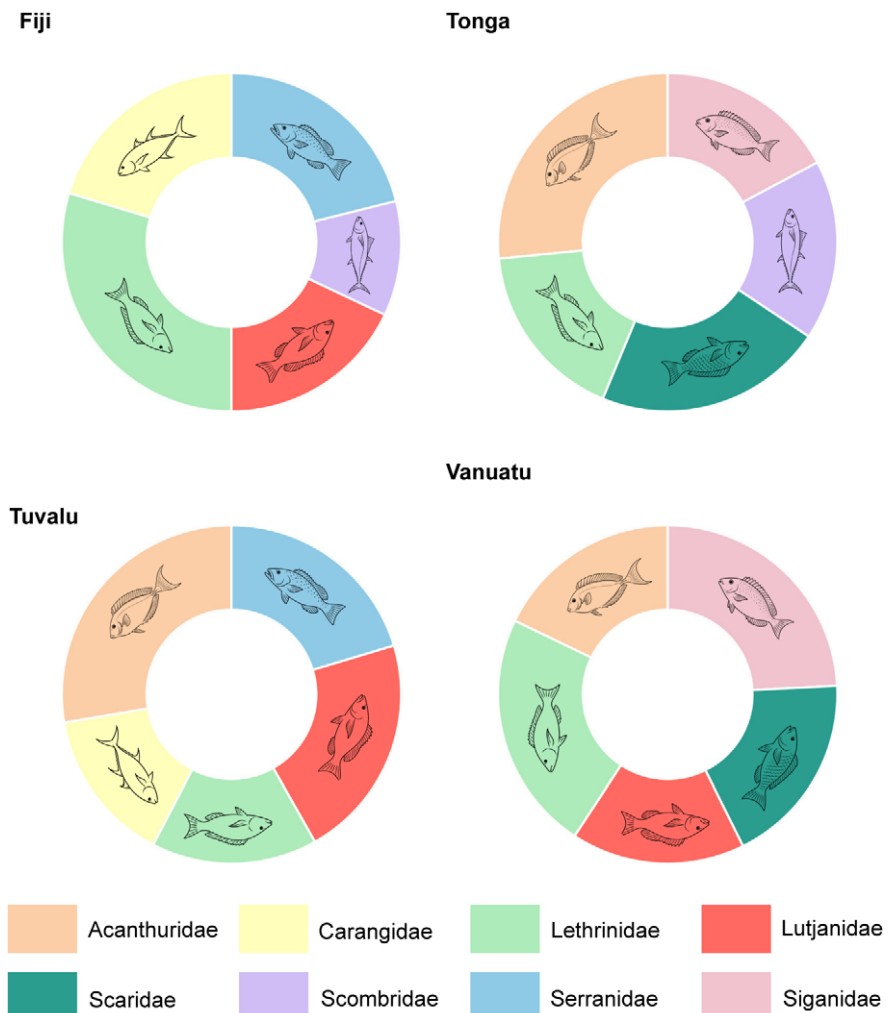

**Figure 2.** Five most popular families identified to be caught within customary fishing grounds by interviewed communities, separated by country. See Supplementary Figure 1 for a Principal Components Analysis of these families.

In Tuvalu, women reported more Mullidae and men Lethrinidae, Lutjanidae, Siganidae and Holocentridae, while Acanthuridae was reported similarly by both. Similarly, in Vanuatu, women reported Mullidae and Serranidae more frequently and men Scaridae, with both genders reporting Lethrinidae. For a full list of gender-disaggregated catch mentions (including taxa without microplastics data), see Supplementary Figure 3.

Lethrinidae ($n$ = 113 mentions), Acanduridae ($n$ = 106 mentions) and Lutjanidae ($n$ = 93 mentions) were identified most commonly on a regional level by fishers. Importantly, for Lethrinidae, the species *Lethrinus harak* (thumbprint emperor) specifically was mentioned 49 times; notably more than any other individual species. The highest ranked families in terms of microplastic loads on a regional level were Scombridae (5.7 ± 1.2 MP/individual), Lethrinidae (1.2 ± 0.1 MP/individual) and Kyphosidae (1.1 ± 0.8 MP/individual). The families with the highest Exposure Index scores regionally were Scombridae, Lethrinidae and Acanthuridae (0.7 ± 0.2 MP/individual) (Table 2).

## Discussion

This study presents a novel, community-informed approach to assessing potential microplastic exposure risks in Pacific Island fisheries and prioritising species for future monitoring efforts. By centring fisher-reported data on catch composition, we identified the fish families most integral to food security in each of the four countries: Fiji, Tonga, Tuvalu and Vanuatu. When paired with microplastic content analysis of these commonly caught fish, these findings offer a locally and regionally relevant risk assessment with regards to which fish may be most likely to transfer microplastics and their associated contaminants to humans. In doing so, this work highlights the vulnerability of communities that rely on coastal fisheries and underscores the importance of integrating local knowledge with scientific pollution monitoring. This combination is vital not only for producing more accurate and context-sensitive assessments but also for shaping waste management and marine policy recommendations that reflect the lived realities and priorities of Pacific communities.

Fishers provided detailed information on the species most commonly caught within customary fishing grounds, reflecting valuable long-term local knowledge and revealing strong patterns that varied by country. Lethrinidae were consistently within the top five families identified across the four countries, while Acanthuridae and Lutjanidae were in the top five reported landings for three countries, highlighting the regional importance of these families. The other common families - Carangidae, Scaridae, Scombridae,

**Table 1.** Table presents Catch Score and MP Score (both normalised values on a 0–1 scale, corresponding to lowest and highest values respectively) and Exposure Index, calculated as a product of these two metrics. Families are ordered from highest to lowest Exposure Index for each country

| Country | Family | Common Name(s) | Catch Score | MP Score | Exposure Index |
|---|---|---|---|---|---|
| Fiji | Scombridae | Mackerels, Tuna | 0.33 | 1.00 | 0.33 |
| | Serranidae | Groupers | 0.69 | 0.43 | 0.30 |
| | Lethrinidae | Emperors | 1.00 | 0.29 | 0.29 |
| | Lutjanidae | Snappers | 0.58 | 0.43 | 0.25 |
| | Scaridae | Parrotfishes | 0.25 | 0.18 | 0.04 |
| | Siganidae | Rabbitfishes | 0.08 | 0.24 | 0.02 |
| | Hemiramphidae | Halfbeaks | 0.14 | 0.10 | 0.01 |
| | Mullidae | Goatfishes | 0.00 | 0.23 | 0.00 |
| | Sphyraenidae | Barracudas | 0.19 | 0.00 | 0.00 |
| Tonga | Siganidae | Rabbitfishes | 0.63 | 0.22 | 0.14 |
| | Scaridae | Parrotfishes | 0.81 | 0.15 | 0.12 |
| | Acanthuridae | Surgeonfishes, Tangs, Unicornfishes | 1.00 | 0.07 | 0.07 |
| | Mugilidae | Mullets | 0.31 | 0.22 | 0.07 |
| | Lethrinidae | Emperors | 0.25 | 0.27 | 0.07 |
| | Mullidae | Goatfishes | 0.31 | 0.16 | 0.05 |
| | Serranidae | Groupers | 0.06 | 0.12 | 0.01 |
| | Lutjanidae | Snappers | 0.69 | 0.00 | 0.00 |
| | Carangidae | Jacks, Trevallies, Pompanos | 0.00 | 1.00 | 0.00 |
| | Labridae | Wrasses | 0.00 | 0.15 | 0.00 |
| Tuvalu | Acanthuridae | Surgeonfishes, Tangs, Unicornfishes | 1.00 | 0.74 | 0.74 |
| | Lutjanidae | Snappers | 0.75 | 0.74 | 0.56 |
| | Mullidae | Goatfishes | 0.23 | 0.99 | 0.23 |
| | Siganidae | Rabbitfishes | 0.19 | 1.00 | 0.19 |
| | Holocentridae | Squirrelfishes, Soldierfishes | 0.19 | 0.90 | 0.17 |
| | Lethrinidae | Emperors | 0.52 | 0.31 | 0.16 |
| | Scaridae | Parrotfishes | 0.15 | 0.45 | 0.07 |
| | Serranidae | Groupers | 0.73 | 0.00 | 0.00 |
| | Sphyraenidae | Barracudas | 0.00 | 0.23 | 0.00 |
| Vanuatu | Mullidae | Goatfishes | 0.57 | 0.40 | 0.23 |
| | Lethrinidae | Emperors | 0.89 | 0.18 | 0.16 |
| | Scaridae | Parrotfishes | 0.78 | 0.20 | 0.15 |
| | Serranidae | Groupers | 0.46 | 0.21 | 0.10 |
| | Hemiramphidae | Halfbeaks | 0.14 | 0.62 | 0.08 |
| | Haemulidae | Grunts, Sweetlips | 0.05 | 1.00 | 0.05 |
| | Carangidae | Jacks, Trevallies, Pompanos | 0.35 | 0.12 | 0.04 |
| | Mugilidae | Mullets | 0.27 | 0.00 | 0.00 |
| | Labridae | Wrasses | 0.19 | 0.00 | 0.00 |
| | Belonidae | Needlefishes | 0.11 | 0.00 | 0.00 |
| | Siganidae | Rabbitfishes | 1.00 | 0.00 | 0.00 |
| | Acanthuridae | Surgeonfishes, Tangs, Unicornfishes | 0.73 | 0.00 | 0.00 |
| | Kyphosidae | Sea chubs, Drummers | 0.00 | 0.00 | 0.00 |

*Note*: Results are presented for each country. To see a complete list of species sampled for microplastics within each family, please refer to Supplementary Table 2, and for the list of taxa fishers identified within each country, please refer to Supplementary Table 3.

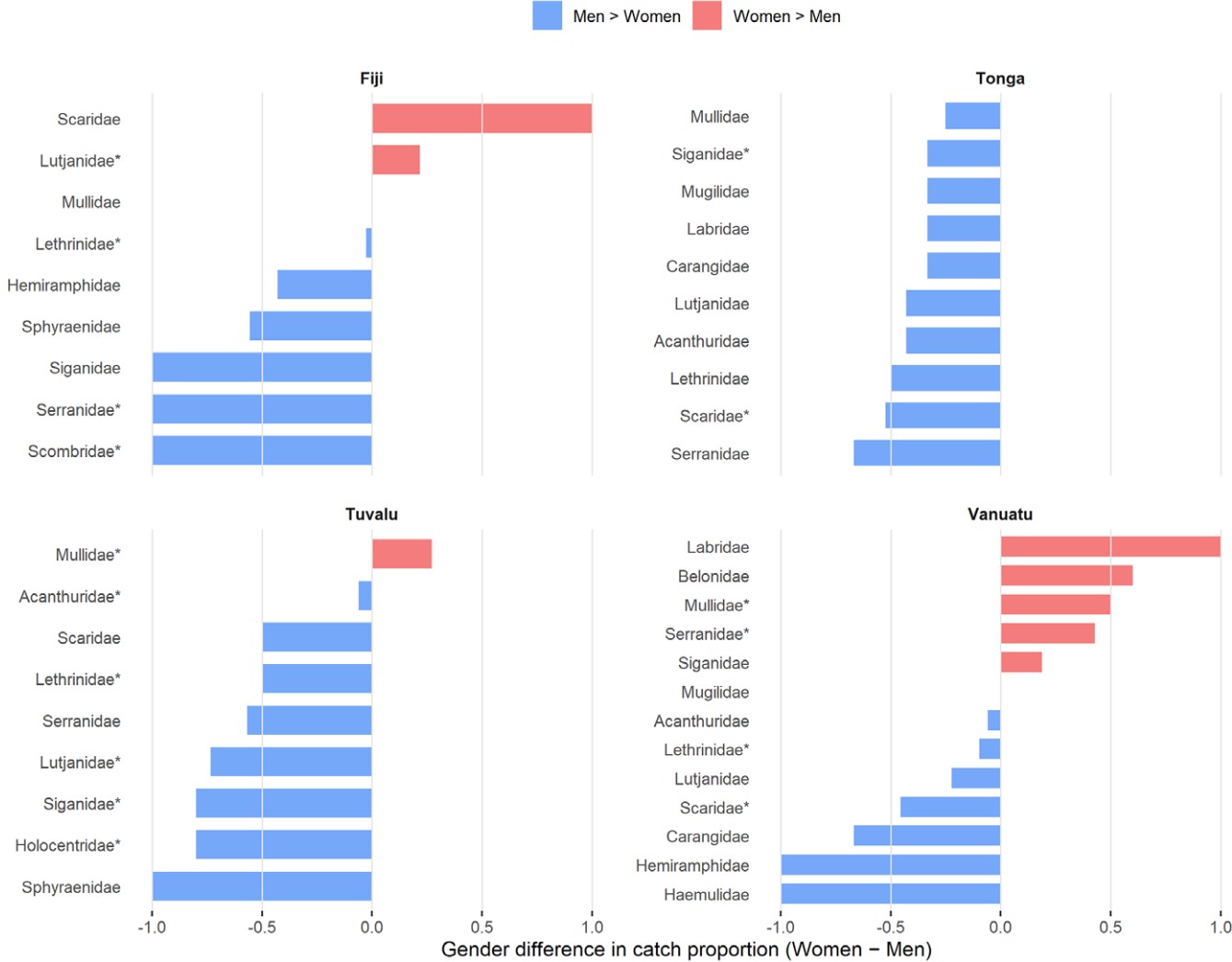

**Figure 3.** Gender disaggregated data on fish families reported as common catch (1.0 = 100% reported by females, 0 = reported consistently among genders; −1.0 = 100% reported by males). Families identified as having the highest Exposure Index (see Table 1; threshold of 0.1) are denoted with an asterisk. Families restricted to those that have microplastics data. For the full list of reported commonly caught species, please refer to Supplementary Figure 3.

Serranidae and Siganidae – all occurred in the top five families of the two countries. Fiji-Tuvalu and Tonga-Vanuatu exhibited more similar commonly reported catches, with four consistent families between each pairing. Differences by country may reflect species ranges, dietary preferences, or local exploitation levels of stocks. These findings strongly reflect country fisheries reports (FAO, 2009; Lee et al., 2020; Tuvalu Fisheries Department, 2022, 2023; Tonga Ministry of Fisheries, 2025; JB Molitaviti pers. comm. 2025), highlighting that local ecological knowledge from fishers is vital for understanding resource use and can complement creel surveys or scientific assessments of fisheries (Peixoto et al., 2022; Silvano et al., 2023). The value of such knowledge is particularly high in areas such as the Pacific Islands, where resources are low and coastal communities are numerous, and local knowledge reflects generations of fisheries experience and practices (Kitolelei et al., 2021). Such data are valuable in understanding food security, as they capture the species most relied upon in daily diets and livelihoods.

Analysis of the Exposure Index, which integrates both microplastic contamination levels and fisher-reported catch frequency, revealed notable differences in exposure profiles among the four Pacific Island countries studied. Importantly, the level of microplastic contamination differs across the four countries (Dehm et al., 2026), with the highest rates in Fiji (75% frequency occurrence, 2.17 ± 1.2 MP/individual in sampled fish; Drova et al., 2025), followed by Tonga (42% frequency occurrence, mean 0.77 ± 0.10 MP/individual; Fe'ao et al., 2025) and Tuvalu (37% frequency occurrence, mean 0.72 ± 0.08 MP/individual; Alefaio et al., 2026), and the lowest rates in Vanuatu (5% frequency occurrence, mean 0.05 ± 0.01 MP/individual; Botleng et al., 2025). In Tuvalu, while microplastic loads were generally moderate, the families Acanthuridae and Lutjanidae had the highest Exposure Index of any families in any country, highlighting the importance of future research focusing on these families. The high Exposure Index values indicate that in Tuvalu, the same families that are most commonly caught are most contaminated with microplastics. Acanthuridae and Lutjanidae have previously been found to vastly dominate night spearfishing and handline catches, respectively, at Funafuti atoll (Moore et al., 2014). In Fiji, high Exposure Index values were driven by the combined catch prominence and contamination of families such as Lethrinidae, Lutjanidae, Scombridae and Serranidae. All four families had both the highest catch scores and microplastic loads, indicating that in Fiji, though not as pronounced as in Tuvalu, fish families central to both food security and income

**Table 2.** Table presents Catch Score and MP Score (both normalised on a 0–1 scale, corresponding to the minimum and maximum observed values, respectively) and the Exposure Index, calculated as a product of the two metrics. Families are ordered from highest to lowest Exposure Index

| Family | Common Name(s) | Catch Score | MP Score | Exposure Index |
|---|---|---|---|---|
| Scombridae | Mackerels, Tuna | 0.43 | 1.00 | 0.43 |
| Lethrinidae | Emperors | 0.97 | 0.21 | 0.21 |
| Acanthuridae | Surgeonfishes, Tangs, Unicornfishes | 1.00 | 0.12 | 0.12 |
| Siganidae | Rabbitfishes | 0.60 | 0.14 | 0.09 |
| Lutjanidae | Snappers | 0.92 | 0.09 | 0.09 |
| Mullidae | Goatfishes | 0.39 | 0.17 | 0.07 |
| Serranidae | Groupers | 0.78 | 0.07 | 0.06 |
| Scaridae | Parrotfishes | 0.59 | 0.08 | 0.05 |
| Carangidae | Jacks, Trevallies, Pompanos | 0.63 | 0.05 | 0.03 |
| Sphyraenidae | Barracudas | 0.12 | 0.15 | 0.02 |
| Holocentridae | Squirrelfishes, Soldierfishes | 0.12 | 0.15 | 0.02 |
| Hemiramphidae | Halfbeaks | 0.11 | 0.12 | 0.01 |
| Kyphosidae | Sea chubs, Drummers | 0.05 | 0.20 | 0.01 |
| Mugilidae | Mullets | 0.32 | 0.02 | 0.01 |
| Haemulidae | Grunts, Sweetlips | 0.05 | 0.06 | 0.00 |
| Labridae | Wrasses | 0.1 | 0.02 | 0.00 |
| Caesionidae | Fusiliers | 0.00 | 0.07 | 0.00 |
| Belonidae | Needlefishes | 0.04 | 0.00 | 0.00 |

*Note*: Results are presented for all four countries combined to provide a regional overview.

generation (supported by data from FAO, 2009) are also those with the highest potential to transfer microplastics to humans. Notably, while most families had good representation across species for microplastic loads in Fiji, the only Scombrid sampled was *Rastrelliger brachysoma*, which was identified by five fishers as being commonly caught, and only three individuals were sampled for microplastics (see Supplementary Table 4). A similar ram-feeding species *Rastrelliger kanagurta* has been reported to filter 19.57 L of water per minute, suggesting high exposure to microplastics in the water column can explain the high microplastic load in this fish (Hamann et al., 2023). Tonga showed quite low Exposure Index scores, indicating that families commonly caught within fishing grounds (e.g. Acanthuridae and Scaridae) were different from those that were most contaminated with microplastics (e.g. Carangidae). Notably, only one fisher in Tonga identified Carangids (specifically 'Jacks' [*Caranx* spp.]) as being commonly caught, but individuals of this family had disproportionately high microplastic loads in Tonga. The importance of Acanthuridae and Scaridae is underscored in the Tonga National Coastal Fisheries Management and Development Plan 2023–2026 (pgs. 31–35). Vanuatu exhibited the lowest average microplastic loads across all families, and, similar to Tonga, had low Exposure Index values, suggesting misalignment between families that are commonly caught and those that are most contaminated with microplastics. The families Mullidae, Lethrinidae and Scaridae emerged as the families of highest interest in Vanuatu.

At the regional level, Lethrinidae stood out as the most significant family, ranking high in both microplastic load and fisher importance, exceeded only by Scombridae, and followed by Acanthuridae. Notably, while microplastics data were available for all four countries for calculating regional means for Lethrinidae

and Acanthuridae with replication of 243 and 49, respectively, Scombridae were only landed and sampled from Fiji, and only three individuals were sampled. This means the regional mean for Scombridae is based on one country and a very small sample size, so it should be interpreted cautiously (see Supplementary Tables 4 and 5). These results underscore the utility of the Exposure Index as a decision-support tool and highlight the potential for species within these families to serve as bioindicators of microplastic exposure in Pacific Island fisheries.

Differences by gender likely reflect differences in habitats where fishing is being conducted or due to different gears being used by women (Kronen and Vunisea, 2009; Ram-Bidesi, 2015). They may also reflect that women are generally responsible for preparing household food (thus potentially perceiving species more commonly consumed in the home as more common), whereas men may be more responsible for selling fish for income (Thomas et al., 2021; McKenzie et al., 2022). Across the four countries, men identified a much larger diversity of catch, but some families, such as Mullidae and Lethrinidae, appeared to be relatively consistently reported by men and women. In most cases, the species with high Exposure Index values were more frequently identified by men, particularly so in Tonga and Tuvalu. Given the documented gendered differences in the species caught and the ultimate destinations of catch (e.g. household consumption vs. income generation; Thomas et al., 2021), ensuring that information on fisheries is collected from both men and women is essential for accurately understanding food security dynamics and for mapping out potential hazard exposure pathways and distribution networks.

The approach used in this study to calculate an Exposure Index offers significant value and provides a robust foundation for future development. By integrating both social relevance and contamination

data, it combines the regularity of fishers' frequency of catch with their associated microplastic loads, offering a comprehensive assessment of real-world exposure risk potential. By focusing on species most relevant to human diets, it highlights those that are most likely to contribute to dietary microplastic intake. This methodology not only provides a practical tool for evaluating potential exposure risk, but also offers an opportunity to support monitoring and management decisions, enabling the identification of priority species for further research, pollution mitigation and public health outreach. However, the findings are subject to certain limitations. While some fish are prepared or consumed with their GITs intact in the PICTs, interpretation of the Exposure Index as an indicator of risk relies on the assumption that microplastics in the GIT reflect either potential translocation of plastic particles and associated chemicals into edible tissues, or possible fish toxicity arising from ingestion exposure (Akhbarizadeh et al., 2019; Barboza et al., 2020). These pathways, however, remain debated and warrant further research (e.g. Jovanović et al., 2018). While the catch data closely align with national fisheries reports, the interviews were conducted with select fishers from only three communities in each country. As such, the results provide an indication of patterns but would benefit from broader coverage and expansion in the future. Microplastic sampling inconsistency across countries, reflecting variations in sampling effort and locations between countries, and limited species representation in some countries, means that certain commonly caught species may not have been sampled for microplastics, creating potential bias for exposure estimates. The use of family-level groupings also obscures species-specific vulnerabilities, limiting fine-scale reasoning. Moreover, the microplastics data were collected at a single point in time and cannot be used to make generalised statements to represent the number of microplastics in all food fish species. Microplastic abundance in field-collected specimens is influenced by temporal and environmental factors, including season, local climatic conditions and hydrodynamics. While the relative differences among taxa (as reflected in the 'MP Score') are expected to be more stable than absolute microplastic loads as they are largely shaped by ecological traits such as feeding mode and habitat use (e.g. Dehm et al., 2026), replication across sites and seasons will be essential for strengthening future applications of the Exposure Index and for providing a more comprehensive assessment of microplastic exposure.

Based on this study, we offer four recommendations for future research:

1. Research should value the contribution of local knowledge to identify species that are crucial to food security.
2. Expand baseline microplastics data using consistent methodologies that build on indices such as this to understand the potential risk of human exposure to microplastics via seafood.
3. Use the exposure index to prioritise indicator species with a combination of high exposure to microplastics (contamination levels) and relevance for food security. The national patterns for Tuvalu in particular highlight Acanthuridae and Lutjanidae as priority taxa for future microplastic monitoring.
4. At a regional level, from this baseline data, fish within the Lethrinidae family, and particularly *L. harak*, emerge as priority species to target in future microplastic surveys.

**Open peer review.** To view the open peer review materials for this article, please visit http://doi.org/10.1017/cft.2026.10027.

**Supplementary material.** The supplementary material for this article can be found at http://doi.org/10.1017/cft.2026.10027.

**Data availability statement.** Data on fisher preferences, species sampled for microplastics and family-level mean microplastic loads are available within the supplementary materials. Data on microplastics at the species level are presented in country-specific data papers (Fiji – Drova et al., 2025; Tonga – Fe'ao et al., 2025; Tuvalu – Alefaio et al., 2026; Vanuatu – Botleng et al., 2025). Species occurrence records are published on the Global Biodiversity Information Facility (Fiji [Drova and Bai, 2023]: https://doi.org/10.15468/4qhgxr; Tonga [Fe'ao, 2025]: https://doi.org/10.15468/5z87zj; Tuvalu [Nivaga et al. 2024]: https://doi.org/10.15468/4jedd8; Vanuatu [Botleng et al. 2024]: https://doi.org/10.15468/tmqxr4).

**Acknowledgements.** We are especially thankful to the coastal communities, community leaders and fishers at all study sites for welcoming us, assisting with sample collection, and sharing their knowledge on local fisheries. This work was strengthened by the technical contributions of government staff, laboratory teams and field assistants who supported sampling and analysis. We gratefully acknowledge the national fisheries departments, ministries and authorities of Fiji, Tonga, Tuvalu and Vanuatu for their valuable institutional support. We also pay tribute to the late Poasi Ngaluafe, former Head of the Fisheries Science Division, Ministry of Fisheries, Tonga, whose early guidance and dedication as the project's original Tongan focal point were invaluable.

**Author contribution.** Conceptualisation: A.K.F., S.K., R.V., K.B., C.M., J.D.; Data Curation: A.K.F., K.B., B.S., J.D.; Formal Analysis: A.K.F.; Investigation: J.V.B., S.A., L.P.K., L.N., E.D., S.F., J.D.; Methodology: A.K.F., R.V., J.D.; Project Administration: A.K.F.; Resources: J.B.M., S.A., L.P.K.; Supervision: A.K.F., R.V., J.D.; Visualisation: A.K.F., J.D.; Writing – Original Draft: A.K.F., J.D.; Writing – Review and Editing: A.K.F., S.K., R.V., B.S., J.V.B., J.B.M., S.A., K.B., C.M., E.D., J.D.

**Financial support.** This research was funded by the Asia-Pacific Network for Global Change Research (grant number: CRRP2022-05MY-Ford, awarded to AKF, grant ID: https://doi.org/10.30852/p.22124).

**Competing interests.** The authors declare no conflicts of interest.

**Ethics statements.** National permits were obtained from the relevant authorities in each country: Fiji (MTA-42/2–3, Ministry of iTaukei Affairs), Tonga (ORG 1/8 v.24, Ministry of Education and Training), Tuvalu (approval from the Director of the Tuvalu Fisheries Authority) and Vanuatu (VAN-ENV-04524). Permission for research was further obtained from relevant provincial and community representatives prior to data collection. For all questionnaires, FPIC was obtained from all individuals, with the right to decline or withdraw at any stage. All data from interviews is anonymous and compiled at the national level. All fish specimens were obtained post-mortem from local fishers.

**Inclusivity statement.** This project was co-designed with fisheries partners in Fiji, Tonga, Tuvalu and Vanuatu from its development. Prior approval was obtained from relevant national and provincial institutions at project initiation. Before fieldwork, the research team co-facilitated national workshops with fisheries partners and invited relevant government ministries, community and provincial representatives to discuss research plans, sampling methods and data-sharing protocols, translated into local languages when required, with opportunities for questions and revision. Following approvals, within communities, information sheets were provided and translated by local research assistants, and FPIC was obtained from all participants prior to interviews and fish sampling, with the option to decline or withdraw at any stage.

Community fishers provided fish for sampling from their fishing grounds, ensuring that sampling reflected real dietary exposure. Interviews were conducted with both women and men to capture gendered catch composition. Local research assistants (mostly recent graduates or current postgraduate students) from each country led interviews and field sampling following training in field protocols, building transferable research skills.

Fisheries partners from each country were invited to a 3-day workshop where preliminary results were discussed and plans developed for collaborative outputs. Technical briefs were drafted collaboratively to be shared with government agencies, and infographic posters on microplastics and fish were

co-designed, then disseminated across all 12 communities. Results were also shared and discussed with communities.

This research strongly aligns with the priorities of local communities by recognising and elevating the value of local fisher knowledge, and using it to interpret potential exposure risks and shape future research. By integrating community insights with scientific analysis to prioritise key food fish for monitoring, the approach ensures that monitoring priorities and management recommendations reflect the lived experience of Pacific Island fishers while supporting communities to safeguard traditional diets and sustain healthy coastal fisheries.

**Declaration of generative AI in scientific writing.** During the preparation of this work, the authors used ChatGPT v.5 in order to improve the readability and language of some sentences in the paper and to draw silhouettes of fish for Figure 2. The tool was in no way used to analyse or draw insights from the data, perform literature searches, or extract any information other than feedback on the writing style based on the provided inputs. The tool was only used to perform minimal changes and provide feedback based on the original input text, where the scientific content of the input sentences remains unchanged. After using this tool, the authors reviewed and edited the content as needed and take full responsibility for the content of the publication.

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
