## [Reviewer Report]

This paper presents a creative and high-quality framework for assessing the real-world risks microplastics pose to communities with high seafood consumption. The methodology is particularly inventive, as it successfully bridges the gap between social consumption data and empirical microplastic load data through the development of the “Exposure Index”. I applaud the authors on such a rigorous publication – well done. I think this will be ready for acceptance after some revisions, which I have highlighted below.

Introduction

The introduction effectively establishes the critical role of coastal fisheries for food security and cultural identity in PICTs. It successfully highlights the disproportionate burden of plastic pollution these nations face despite their minimal global contribution to waste. The rationale for integrating local ecological knowledge to identify exposure pathways is well-supported and establishes the study’s novelty. Well done!

Methods

The methods section is where I believe the most clarification is required. Currently it lacks the granular detail necessary for full reproducibility and should be expanded in the following areas:

Interviews: While the text mentions “questionnaire-guided interviews,” providing the specific questions asked of the 110 fishers would be highly beneficial. Understanding whether the “top ten species” were identified through free-listing or prompted from a list is essential for evaluating potential bias. I also think this would be very helpful if other scientists were hoping to replicate your work in other regions.

Qualitative data analysis: The manuscript should explicitly state how interview data were analysed. For example, it is unclear if a thematic analysis was used to interpret fisher insights or if the data were purely treated as frequency counts for the “Catch Score”. The interview data in general needs a lot more expansion and clarification.

Microplastic sampling: More detail is required regarding the QA/QC measures. While “sterile conditions” and “FTIR polymer analysis” are mentioned, providing the specific number of procedural blanks or the recovery rates for the digestion process would improve technical rigor.

The text lacks immediate clarity on species-level sample sizes. For instance, the regional findings for Scombridae are based on a very small sample of only three individuals. A summary table within the main text (rather than just the supplementary materials) showing the number of fish analysed per species would prevent the assumption that sample sizes might be as low as one per species. I think Table S4 should be moved to the results or methods.

Results

The results section provides a strong quantitative overview of the Exposure Index but misses an opportunity to showcase the qualitative richness of the 110 interviews.

Beyond the frequency of catch, the results could be enhanced by including fishers' observations on habitat changes, seasonal shifts, or specific feeding behaviours they may have reported. I am unsure what sort of data was collected during the interviews (another benefit of including the interview questions) but I think there is a missed opportunity here to report some potentially really novel results.

Figure 2 – I would add silhouettes of the different species that you referring to in the donut graphs – this would make it more visually appealing but also help readers to associate family/species names when possible.

Discussion

The discussion provides a robust interpretation of the results but would benefit from some expansion/clarification in areas.

The authors correctly note in the limitations that microplastic abundance is influenced by local climate, tides, and seasons. The discussion should further explore how the “single point in time” sampling might affect the dynamic nature of the Exposure Index.

While gendered differences in catch are documented (super interesting!), the discussion could more explicitly link these to public health outcomes. For example, if women are primarily responsible for preparing food, they may face different exposure risks through handling or the consumption of specific “non-market” species that men do not target for sale.

Minor edits:

Referencing error in the caption for Figure 3, which refers to a “Table 3” for identifying high-risk families; however, the provided tables are numbered Table 1 and Table 2?

---

## [Reviewer Report]

The authors are presenting a study based on the integration of local knowledge with empirical data by combining social science and experimental data. While the approach is of interest I did find the manuscript to be lacking of focus and would benefit from a major restructure. As a result I would propose major revisions with resubmission. More comments below:

- More details are needed in the method section regarding the questionnaire. Dates are missing, how did you develop the questionnaires, did you follow any specific protocol?

- Could you add a copy of the questionnaire in the SI document?

- More details are needed regarding the microplastics work. What were the contamination control procedures, why using H2O2 and such a high temperature of 60 degrees Celsius. Did you follow any specific method/protocol?

- Same question for the FTIR work. Which instrument did you use with which libraries, what about describing particles, is all the info in the other papers in prep?

- Papers in prep should still be listed in reference list

- I am not convinced also with the exposure index. Unless I am mistaken all GITs are removed before consumption. I would focus the study on the role of these species as sentinels to understand microplastics distribution rather than focusing on the notion of harm which would be more focused on particles small enough to pass the cell barriers like small micro to nano.

- For the definition of microplastics I would also add the minimum size to avoid overlap with nanoplastics

---

## [Reviewer Report]

Well done to the authors on addressing all of my queries. In regards to their comments on figure 2, I do think this is improved (although I understand the issue with some fish looking similar). I am happy with how it is, or another option for simplicity could be to put the outline of the fish only next to the colours in the legend, if you prefer this. I will leave this to the author team. Otherwise, well done, and I am happy to see this paper accepted.

---

## [Reviewer Report]

The authors have attended to all comments from the reviewers. I am satisfied with the responses as provided by the authors.